# Predicting marsh vulnerability to sea-level rise using Holocene relative sea-level data

Benjamin P. Horton[1,2], Ian Shennan [3], Sarah L. Bradley[4], Niamh Cahill[5], Matthew Kirwan [6], Robert E. Kopp [7,8] & Timothy A. Shaw[1]

Tidal marshes rank among Earth's vulnerable ecosystems, which will retreat if future rates of relative sea-level rise (RSLR) exceed marshes' ability to accrete vertically. Here, we assess the limits to marsh vulnerability by analyzing >780 Holocene reconstructions of tidal marsh evolution in Great Britain. These reconstructions include both transgressive (tidal marsh retreat) and regressive (tidal marsh expansion) contacts. The probability of a marsh retreat was conditional upon Holocene rates of RSLR, which varied between −7.7 and 15.2 mm/yr. Holocene records indicate that marshes are nine times more likely to retreat than expand when RSLR rates are ≥7.1 mm/yr. Coupling estimated probabilities of marsh retreat with projections of future RSLR suggests a major risk of tidal marsh loss in the twenty-first century. All of Great Britain has a >80% probability of a marsh retreat under Representative Concentration Pathway (RCP) 8.5 by 2100, with areas of southern and eastern England achieving this probability by 2040.

[1] Asian School of the Environment, Nanyang Technological University, Singapore 639798, Singapore. [2] Earth Observatory of Singapore, Nanyang Technological University, Singapore 639798, Singapore. [3] Department of Geography, Durham University, Durham DH1 3LE, UK. [4] Department of Geoscience and Remote Sensing, Delft University of Technology, Delft 2628, The Netherlands. [5] School of Mathematics and Statistics, University College Dublin, Dublin 4, Ireland. [6] Virginia Institute of Marine Science, College of William and Mary, Gloucester Point, VA 23062, USA. [7] Institute of Earth, Ocean, and Atmospheric Sciences, Rutgers University, New Brunswick, NJ 08901, USA. [8] Department of Earth and Planetary Sciences, Rutgers University, Piscataway, NJ 08854, USA. Correspondence and requests for materials should be addressed to B.P.H. (email: bphorton@ntu.edu.sg)

Tidal marshes are vulnerable to relative sea-level rise (RSLR), because they occupy a narrow elevation range, where marshes retreat and convert to tidal flat, tidal lagoon, or open water if inundated excessively[1–3]. But regional and global models differ in their simulations of the future ability of marshes to maintain their elevation with respect to the tidal frame[4]. Some landscape models predict up to an 80% decrease in global tidal marsh area by 2100[5], with substantial marsh loss even when RSLR rates are less than 8 mm/yr[6,7]. By contrast, other simulation studies suggest that, through biophysical feedback and inland marsh migration, marsh resilience to retreat is possible at RSLR rates in excess of 10 mm/yr[2,4,8,9].

The compilation of empirical data for tidal marsh vulnerability is essential to addressing disparities across these simulation studies. Marshes respond to RSLR in part by building soil elevation, and vertical sediment accretion data are available for many marshes in North America and Europe. Some meta-analyses suggest that marshes are generally resilient to modern rates of RSLR, because they build vertically at rates that are similar to or exceed RSLR[3,4], whereas others suggest that submergence is already taking place[10]. The outcomes of tidal marsh vulnerability often reflect site-specific differences in the physical and biological setting[1,11–13]. But comparing current accretion rates to future rates of RSLR may be problematic for three reasons. First, accretion rates tend to increase with flooding duration so that marshes may accrete faster under accelerated RSLR[4,14]. Therefore, simple comparisons between current vertical accretion and future RSLR may overestimate marsh vulnerability[4]. Second, twentieth and early twenty-first century rates of RSLR varied from −2.5 to 3.7 mm/yr (5–95th percentile range among tide-gauge sites[15]), and are dwarfed by potential future rise, which under high forcing and unfavorable ice sheet dynamics could exceed 2 m by 2100 (i.e., a century-average rate of 20 mm/yr) in many locations[16]. Indeed, in Louisiana, a comparison between rates of RSLR, which are locally enhanced by sediment compaction to >12 mm/yr, and vertical accretion illustrates over 50% of the tidal marshes are not keeping pace with sea level[10]. Finally, lateral erosion threatens marshes even when they are accreting vertically in pace with RSLR[17,18]. Thus, additional measures of tidal marsh response are needed to accurately forecast marsh vulnerability to RSLR.

Here we assess the limits to marsh vulnerability for Great Britain by analyzing reconstructions of tidal marsh retreat and expansion during the Holocene. The tidal marshes of Great Britain have expanded, remained static, and retreated while RSLR varied between −7.7 and 15.2 mm/yr (Fig. 1), primarily because of the interplay between global ice-volume changes and regional isostatic processes[19]. We can, therefore, analyze the trends in the Holocene data to explore the limits to marsh vulnerability with rates of RSLR greater than twentieth and early twenty-first century rates. Great Britain has the largest Holocene sea-level database in the world[20,21] and has 20 years of integration between data collectors and the glacial isostatic adjustment (GIA) modeling community[19,22,23]. Local relative sea-level (RSL) records have been reconstructed from sea-level index points, which each provide a discrete reconstruction from a single point in time and space[20]. We employ a GIA model[19] to determine the rates of RSLR for each index point. While sea-level index points are most commonly used to assess past RSL[24], here we make use of additional associated information to assess the resilience of tidal marshes, or lack thereof, to past rates of RSLR. Sea-level tendency[25] describes the increase or decrease in marine influence recorded by an index point, as indicated by a change in tidal marsh sediment stratigraphy or a transgressive or regressive contact[25]. Transgressive contacts, describing changes in depositional environment from tidal marsh to tidal flat (tidal marsh

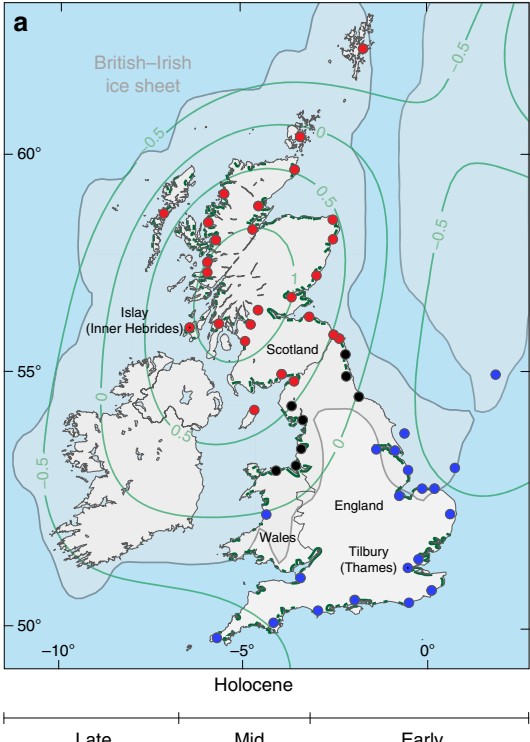

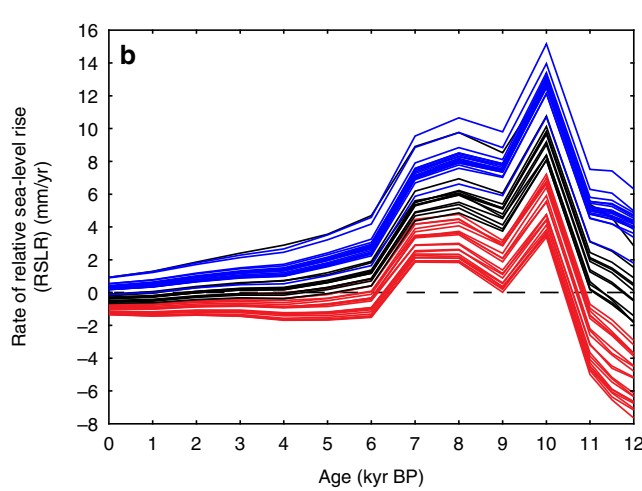

**Fig. 1** The Great British Holocene relative sea-level database. **a** Location of the 54 regions used to group individual sea-level index points. Approximate spatial extent (in light blue) of the British–Irish ice sheet (BIIS) at the last glacial maximum (21,500 cal. yrs. BP), redrawn from ref. [19] (Copyright© 2011 John Wiley & Sons, Ltd). Contours represent the predicted present-day rate of land-level change, where relative uplift is positive, subsidence is negative (mm/yr) using the model from ref. [19]. Current areas of tidal marshes are shown (in green) following ref. [51]; **b** Holocene rates of relative sea-level rise (RSLR) for 54 locations (Supplementary Table 3) of the Great British database of sea-level index points using the Bradley GIA model[19] (Methods). The red dots and lines are sites that are located close to the center of BIIS loading; black dots and lines are sites at the margin of the BIIS; and blue dots and lines are sites distal to the BIIS

retreat), have a positive tendency (increasing marine influence). Regressive contacts reflect a negative tendency (decreasing marine influence) and describe the replacement of a tidal flat by a tidal marsh deposit (tidal marsh expansion). Stratigraphic evidence of a positive tendency include a change from freshwater peat to a tidal marsh deposit, or a change in microfossil assemblages

indicating an increasing marine influence, and vice versa for negative tendencies. Based on the Holocene relationship between GIA-modeled rates of RSLR and sea-level tendency, we estimate the probability of a positive tendency conditional upon different rates of RSLR. This probability distribution is used to predict the future timescale of marsh vulnerability in Great Britain, by coupling it with local projections of future RSLR under different emission trajectories.

## Results

**Great British Holocene relative sea-level database.** We compiled the RSL data for 54 regions (Fig. 1a) from the Great British Holocene RSL database and integrated with GIA modeling predictions of rates of RSL change (Fig. 1b; Methods). The RSL data and GIA predictions can be subdivided into regions close to (red), at the margins of (black), and distal to (blue) the center of the last glacial maximum British–Irish ice sheet. Sea-level index points in regions of Scotland, close to the center of ice loading, record a non-monotonic pattern, showing deglacial RSL fall during the early Holocene (−7.7 to −0.7 mm/yr), before a rise throughout the mid-Holocene (0.0–6.0 mm/yr) to create a highstand, which was followed by RSL fall to present (−1.7 to 0.0 mm/yr). In middle Great Britain (NE and NW England), at regions closer to the margins of the last glacial maximum ice limit, there is a transition from sites with a small or minor mid-Holocene highstand to sites where RSL is below present throughout the Holocene. Regions along the southern coasts of Great Britain illustrate the characteristic pattern of RSL change of sites distal to the main center of ice loading. The characteristic RSL trend here is a gradual rise over the Holocene toward modern

sea level, with rates of RSLR higher in the early Holocene (15.2–3.1 mm/yr) than in the mid-Holocene (10.7–5.7 mm/yr) and late Holocene (4.6–0.0 mm/yr).

**Sea-level tendency.** The Great British Holocene RSL database of sea-level tendencies has an approximately even distribution of index points with positive ($n = 403$) and negative ($n = 360$) tendencies (Supplementary Fig. 1). The database also includes tidal marsh index points that show no tendency ($n = 19$), indicating the marsh is stable and keeping pace with RSLR. We take only those index points from our database that come from gradual contacts between sediment layers (i.e., 781 index points from the original 1097; Supplementary Fig. 2), reducing the range of RSLR rates to −5.5–10.0 mm/yr.

The rates of RSLR for index points that have positive, negative, and no tendencies are between −0.5 and 10.0 mm/yr, −5.5 and 7.0 mm/yr, and −1.0 and 7.5 mm/yr, respectively (Fig. 2a). The proportion of positive, negative, and no tendencies for each RSLR rate shows only negative tendencies (marsh expansion) for RSL between −1.5 and −5.5 mm/yr, only positive tendencies (marsh retreat) for RSL between 8.0 and 10.0 mm/yr, and a general increase in the proportion of positive tendencies for RSL between 0 and 7.5 mm/yr (Fig. 2b). The latter observation, a range in which some sites record marsh retreat and others record marsh expansion, is consistent with observations from across Great Britain under historical RSLR rates[26].

**Statistical model of sea-level tendency.** To estimate the probability of a positive tendency conditional upon rates of RSLR in the Great British Holocene RSL database, we convert the

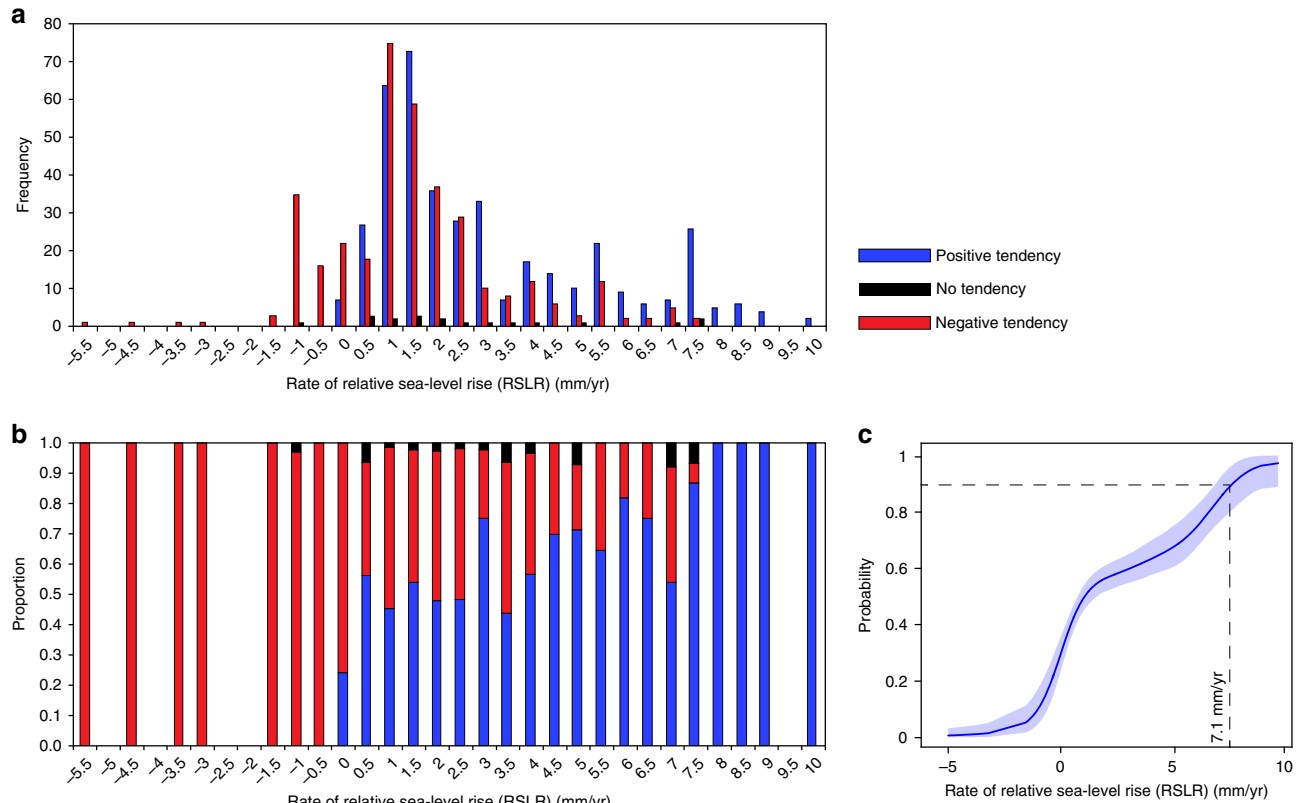

**Fig. 2** Rates of relative sea-level rise for positive, negative, and no tendency sea-level tendencies. **a** Histogram of number of positive, negative, and no tendency sea-level tendencies for rates of relative sea-level rise (RSLR; 0.5 mm/yr bins); **b** Proportion of positive, negative, and no tendency sea-level index points, recording marsh retreat, marsh expansion, and marsh keeping pace with RSLR, respectively, for rates of RSLR (0.5 mm/yr bins); **c** Probabilities of having positive sea-level tendency associated with different rates of Holocene RSLR. Note: no index points in the data set occur outside of the range shown

tendency data into a binary response variable (negative and no tendency = 0, positive tendency = 1) and treat them as having a Bernoulli distribution. The probabilities parameterizing the distribution are estimated by modeling their functional relationship with the RSLR rates (Methods). We summarize this distribution using the probabilities of having positive sea-level tendency associated with different rates of Holocene RSLR (Fig. 2c). When rates of RSLR are ≥7.1 mm/yr, the probability of a positive tendency increases to ~90% (95% uncertainty interval (UI): 80–99%), making the tidal marsh nine times more vulnerable to retreat and conversion to tidal flat than marsh expansion or remaining stable. Conversely, when RSLR rates in the database are ≤−0.2 mm/yr, the probability of having a positive tendency decreases to ~10% (95% UI: 5–27%); therefore, a marsh is very likely to expand or remain unchanged under falling RSL (Fig. 2c).

Modern observations from the southern coasts of Great Britain show that frequently flooded, low-elevation marshes typically build elevation at a rate of 4–8 mm/yr and high-elevation marshes build at rates less than 3 mm/yr[27,28]. Comparison of these modern observations and our analysis of Holocene data suggest that when RSLR exceeds 7.1 mm/yr, at least some marshes would begin to retreat (positive tendencies) and that conversion from high marsh vegetation to terrestrial environments (negative tendencies) would be highly unlikely. Since expansion of marshes over tidal flats (another source of negative tendencies) is unlikely except when RSL is falling or slowing rising, modern observations of salt marsh accretion are at least generally consistent with our finding that marsh retreat in the Holocene has been far more common than marsh expansion under rapid RSLR. Marsh area changes in the rapidly subsiding Mississippi Delta region may serve as an important modern analog. Across the Louisiana Coast, where the mean rate of RSLR is 12.8 mm/yr[10], land loss (1788 square miles, 1932–2010) is ~17 times greater than areas of land gain (104 square miles, 1932–2010)[29].

**Sea-level rise projections for Great Britain**. We generated probabilistic projections of future RSLR following ref. [15] (Methods) at decadal intervals for locations of tidal marsh of Great Britain under the high-emission Representative Concentration Pathway (RCP) 8.5 and low-emission RCP 2.6 trajectories. Projected RSLR varies across Great Britain predominately due to continuing GIA[19], but also due to the static-equilibrium fingerprint of transferring mass from Greenland to the ocean[30], atmosphere/ocean dynamics[31], and local processes such as compaction[32].

The Thames marshes are in an area of GIA subsidence. Under the RCP 8.5 projections, RSL at Tilbury, located within the Thames Estuary, very likely ($P = 0.90$) rises by 23–123 cm between 2000 and 2100, with rates of RSLR of 3–7 mm/yr between 2010 and 2030, 3–11 mm/yr between 2030 and 2050, and 1–18 mm/yr between 2080 and 2100 (Supplementary Table 1). Because sea level responds slowly to climate forcing[33], projected rates of RSLR before 2050 are only weakly reduced under RCP 2.6. But by 2100 there are notable reductions, with a very likely RSLR of 7–83 cm between 2000 and 2100, and rates of −1–11 mm/yr between 2080 and 2100 (Supplementary Table 2).

In the numerous tidal marshes in regions near the center of relative uplift over Scotland, for example Islay, the Inner Hebrides (Supplementary Tables 1 and 2), a very likely rise of 1–96 cm between 2000 and 2100 is projected under RCP 8.5, and −12–63 cm under RCP 2.6. GIA uplift reduced the very likely rates of RSLR under RCP 8.5 for the Inner Hebrides to 1–5 mm/yr between 2010 and 2030, 0–9 mm/yr between 2030 and 2050, and −1–15 mm/yr between 2080 and 2100.

**Responses of tidal marshes to future sea-level rise**. We couple the local projections of RSLR under RCP 8.5 and 2.6 trajectories (Supplementary Tables 1 and 2) with the probability of having positive tendencies associated with different rates of Holocene RSLR (Fig. 2c) to project the timescale of marsh vulnerability in Great Britain (Methods). We produce maps of locations of tidal marsh of Great Britain showing: (1) the year of probability $P > 0.8$ for a positive sea-level tendency (Fig. 3); and (2) the probability of a positive sea-level tendency for 2020, 2040, and 2090 (Supplementary Figs. 4 and 5) under high-emission RCP 8.5 and low-emission RCP 2.6 trajectories.

Nearly all locations of tidal marsh in Great Britain have a >80% probability of a positive tendency (marsh retreat) under RCP 8.5 by 2100, with areas of southern and eastern England (areas of GIA subsidence) achieving this probability by 2040 (Fig. 3a). Throughout Scotland and northwestern England (areas of GIA uplift or negligible land-level change), reducing emissions to RCP 2.6 is sufficient to maintain a >20% probability of a negative or no tendency (marsh expansion or remaining unchanged) for at least the next two centuries (Fig. 3b). However, there remains a >80% probability of a positive tendency within the twenty-second century along the southeastern and eastern coasts of England. Our projections do not account for the elevated probability of Antarctic ice sheet contributions close to ~1 m in RCP 8.5 indicated by some recent modeling studies[34]; integrating such a possibility would further increase the probability of a positive tendency throughout Great Britain in the second half of the twenty-first century and beyond, particularly under RCP 8.5[16].

The high rates of RSLR experienced in much of Great Britain during the early Holocene will become increasingly common in the twenty-first century, with ensuing consequences for tidal marsh environments. Our predicted timescales of marsh vulnerability suggest a nearly inevitable loss of these ecologically and economically important coastal landforms[35] in the twenty-first century and beyond for rapid RSLR scenarios.

## Methods
**Great British Holocene relative sea-level database**. The index points from Holocene RSL database for Great Britain are derived from stratigraphic sequences that record tidal marsh retreat and advance between peat-dominated freshwater ecosystems and increasingly minerogenic tidal marsh, tidal flat (the term tidal flat includes a range of unvegetated, intertidal environments with a range of minerogenic grain sizes, including clay, silt, and sand), and subtidal deposits. The database includes tidal marshes that evolved in different physiographic conditions, climates, substrates, and salinities, overcoming some of the limitations of comparing past, present, and future environmental conditions[36]. It should also be noted that landward marsh migration was possible during the Holocene. Dykes typically prevent modern British tidal marshes from migrating inland[26].

The Great British Holocene RSL database is derived from 54 regions based on availability of data and distance from the center of the British–Irish ice sheet (Supplementary Table 3). The database includes over 80 fields of information for each index point[20], with a subset of the fields relevant to determine tidal marsh vulnerability: (1) Location—geographical co-ordinates of the site from which the index point was collected; (2) Age—estimated using radiocarbon ($^{14}$C) dating of organic material contained within former tidal marshes and calibrated to sidereal years; (3) Tendency—describes the increase or decrease in marine influence recorded by the index point. Tendency does not imply the operation of any vertical movement of sea level[37]; and (4) Lithology above and below the stratigraphic contact. Index points with positive tendencies come from the gradual transgressive contact between tidal marsh and the overlying tidal flat unit, or a change from freshwater peat to a tidal marsh deposit, or a change in microfossil assemblages indicating an increasing marine influence. Therefore, positive tendencies represent marsh retreat. We exclude samples where the contact is erosional as the age is only a minimum age for the erosion event, and we do not know the duration of the hiatus. A similar methodology was applied to negative tendencies. Index points on regressive contacts reflect a negative tendency and describe the gradual replacement of a tidal flat deposit by a tidal marsh deposit (tidal marsh expansion). Index points ($n = 19$) from tidal marsh peat, overlain by tidal flat deposits, but not directly from the transgressive contact and with no evidence of an increasing marine influence in either the lithology or microfossil assemblages (if present) are classed as no tendency, and indicate the marsh is stable and keeping pace with RSLR.

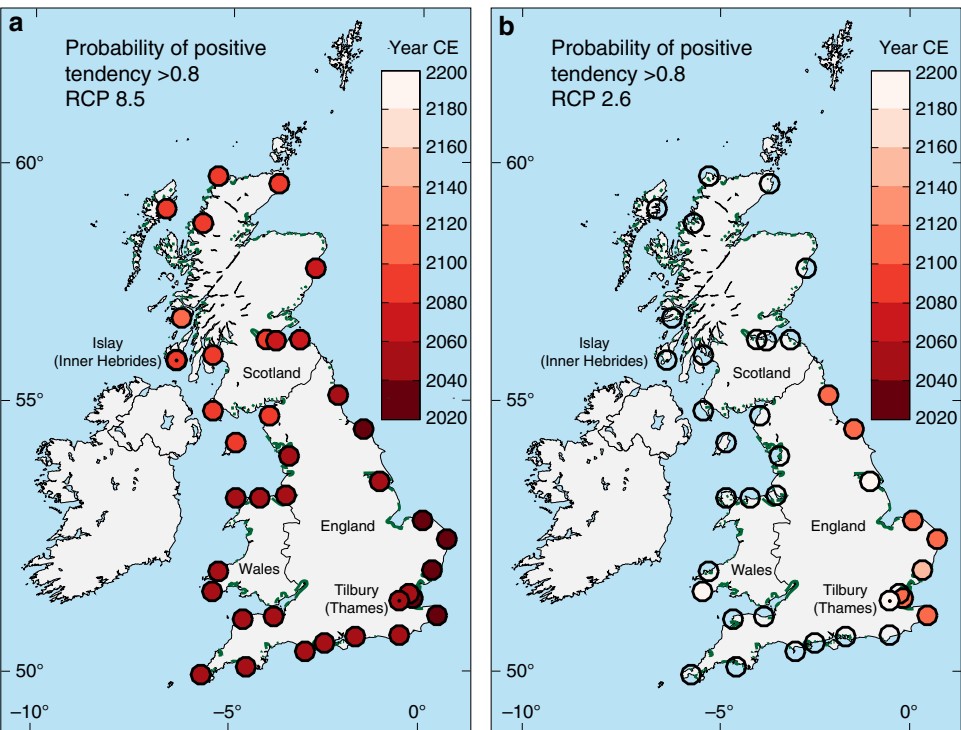

**Fig. 3** Probability for a positive sea-level tendency under different emission pathways. Maps of selected locations in Great Britain showing the year of probability $P > 0.8$ for a positive sea-level tendency under **a** high-emission Representative Concentration Pathway (RCP) 8.5 and **b** low-emission RCP 2.6 pathways. Current areas of tidal marshes (in green) following ref. [51]. Tilbury and Islay are highlighted (black dots in circles)

The index points cover the time period 0–12,000 calibrated years before present (cal. yrs. BP). Most of the data are distributed temporally between 3000 and 8000 cal. yrs. BP (Supplementary Fig. 1). RSL rates between 0 and +3 mm/yr occur more frequently during this temporal period (Fig. 1b). Therefore, we examine the proportion of positive, negative, and no tendencies for each RSL rate (Fig. 2b).

**Example of a positive and negative sea-level tendency**. Supplementary Fig. 2 depicts the interpretation of lithological and microfossil sea-level indicators from core 95/3 at Warkworth, Northumberland[38] to produce two sea-level index points from regressive (negative sea-level tendency) and transgressive (positive sea-level tendency) contacts. A thin clay unit lies between a basal till unit and peat (Supplementary Fig. 2d). Estuarine and low tidal marsh foraminifera in the clay (e.g., *Miliammina fusca*) indicate deposition in a tidal flat environment (Supplementary Fig. 2e). In the peat, pollen assemblages, characterized by herbaceous taxa (*Chenopodiaceae, Cyperaceae,* and *Gramineae*) and tree and shrub taxa (*Betula, Pinus, Quercus,* and *Corylus*) indicate deposition in a high tidal marsh environment. This is corroborated by an abundance of high tidal marsh foraminifera (e.g., *Jadammina macrescens*). Together, these inferences reflect a decrease in marine influence and mark a negative sea-level tendency (regressive contact), which was radiocarbon dated to 8439–8956 cal. yrs. BP.

Overlying the peat, within a second clay unit, estuarine and low-salt marsh foraminifera and dinoflagellate cysts (e.g., *Spiniferites*) indicate tidal flat deposition. These inferences reflect an increase in marine influence and a positive sea-level tendency (transgressive contact), which was radiocarbon dated to 8501–8959 cal. yrs. BP.

The sea-level index points from Warkworth and other locations in Northumberland combine to show Holocene RSL rise from −5 m at 8500 cal. yrs. BP to 0 m at 4300 cal. yrs. BP and culminating in a mid-Holocene highstand ~0.2 m above present[20,38]. This pattern conforms to glacial isostatic adjustment predictions for an area within the limits of ice advance at the last glacial maximum[19,23]. Regional scatter of index points reflects the influence of local-scale processes such as tidal-range change and sediment consolidation.

**Glacial isostatic adjustment model**. We employ a glacial isostatic adjustment (GIA) model[19] to determine the rates of RSLR for each index point of the database, which records tidal marsh expansion or retreat. The key parameters of the GIA model[19] (referred to as the Bradley) are (1) a reconstruction of the Late Quaternary ice change commencing at ~120,000 yrs. BP; (2) an Earth model to reproduce the solid Earth deformation resulting from surface mass redistribution between ice sheets and oceans; and (3) a model of RSL change to calculate the redistribution of ocean mass, which includes the influence of GIA-induced changes in Earth rotation and shoreline migration[39,40].

The Bradley model combined two regional ice sheet reconstructions; one for the British ice sheet[41] and one for Irish ice sheet[42] with a global GIA model. The spatial and temporal record of the British–Irish ice sheet was developed using geomorphological evidence with the maximum vertical height delimited by Scottish trimline data[43,44]. Using the sea-level index point database from both Great Britain and Ireland and GPS data, chi-squared analysis ($\chi^2$) was used to determine the optimal range of earth model parameters for the Bradley model (Supplementary Table 4).

The GIA model predicts RSL predictions for the exact location of each sea-level index point. However, as the temporal resolution of the GIA model is 1000 yrs. to calculate the RSL at the median age of each sea-level index point, we use linear interpolation. Using the predicted RSL at each sea-level point, the rates were then calculated over a 200 yr. (±100 yrs.) interval (Supplementary Fig. 3).

**Statistical model**. The tendency data are coded as binary (negative tendency = 0, positive tendency = 1) and we assume the data $y$ follow a Bernoulli distribution:

$$y_i \sim \text{Bernoulli}\,(p_i), \text{for } i = 1, \dots N,$$

where, $N$ is the total number of observations and $p_i$ is the probability that observation $i$ has a positive tendency. The $p_i$ were estimated by modeling their functional relationship with RSLR rates (denoted $x_i$). A flexible cubic penalized B-spline[45] function was used to model the logit transformed $p_i$'s to insure the probabilities where constrained 0 and 1,

$$\text{logit}(\mathbf{p}) = \sum_{k=1}^{K} \mathbf{b}_k(\mathbf{x})\alpha_k,$$

where $b_k$ is the $k$th cubic B-spline evaluated at $\mathbf{x}$, $K$ is the total number of cubic B-splines, and $\alpha_k$ refers to spline coefficient $k$. The first-order differences of the spline coefficients were penalized to ensure smoothness of the fitted curve as follows:

$$\alpha_k - \alpha_{k-1} \sim \text{N}\big(0, \sigma_\alpha^2\big),$$

where $\sigma_\alpha^2$ determines the extent of the smoothing, a smaller variance corresponds to a smoother trend. A further constraint was imposed on the coefficients so that their differences could not be less than zero, therefore, insuring the resulting trend increased monotonically. The model was fitted in a Bayesian framework and posterior samples of $p_i$ were obtained using a Markov chain Monte Carlo (MCMC) algorithm, implemented in software packages R[46] and JAGS[47] (just another gibbs

sampler). The posterior samples form a posterior distribution for $p_i$ from which we obtained point estimates for the probabilities of positive tendencies with uncertainty.

**Sea-level projections.** Several data sources are available to inform sea-level projections[48–50]. Here, sea-level rise projections follow the framework of ref. [15], which synthesizes probability distributions for a variety of contributing factors including land-ice changes, ocean thermal expansion, atmosphere/ocean dynamics, land water storage, and background geological processes such as GIA. Regional variability in the projections arise from the static-equilibrium fingerprints of land-ice changes, from atmosphere/ocean dynamics, and from non-climatic background processes (including GIA). We generated sea-level projections for tide-gauge locations that are near tidal marshes of Great Britain using 10,000 Monte Carlo samples from the joint probability distribution of different contributing factors (Supplementary Tables 1 and 2). To determine the probability of a positive tendency, for each Monte Carlo sample at each point in time, we take the mean estimate of the probability of a positive tendency conditional on the cumulative maximum of the 20-year average rate of change from the constrained P-spline, then take the expectation of these probabilities across Monte Carlo samples.

**Data availability.** The Great British Holocene relative sea-level database is available from the corresponding author on request. All other data supporting the findings of this study are available within the paper (and its supplementary information files).

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

## Acknowledgements

This research is supported by the National Research Foundation Singapore and the Singapore Ministry of Education under the Research Centres of Excellence initiative. R.E.K. and B.P.H. were supported by the National Science Foundation ARC-1203415 and OCE-1458904 and the Community Foundation of New Jersey, and David and Arlene McGlade. S.L.B. acknowledges support the European Research Council ERC-StG-678145-CoupledIceClim. M.K. was supported by the National Science Foundation DEB-1237733, OCE-1426981, and EAR-1529245. This paper is a contribution to PALSEA2 (Palaoe-Constraints on Sea-Level Rise) and International Geoscience Programme (IGCP) Project 639 "Sea Level Change from Minutes to Millennia." This is Earth Observatory of Singapore contribution 202.

## Author contributions

B.P.H. designed and oversaw all aspects of the research and took the lead on writing the manuscript. I.S. led the construction of the Great British Holocene relative sea-level database. S.L.B. developed the glacial isostatic adjustment model. N.C. applied a statistical model to estimate the probability of a positive tendency conditional upon rates of sea-level rise in the Great British Holocene RSL database. R.E.K. generated probabilistic projections of future relative sea-level rise. Selected portions of the manuscript or supplement were written by B.P.H., I.S., S.L.B., N.C., M.K., R.E.K. and T.A.S. All authors reviewed the manuscript.

## Additional information

**Competing interests:** The authors declare no competing interests.

