## [Peer Review File · Nature Communications]

Reviewers' comments:

Reviewer #1 (Remarks to the Author):

The authors use stratigraphic data to suggest marsh loss in Great Britain is inevitable should the future rate of RSLR exceed 7.3 mm/yr. They base this forecast on a probability distribution of positive tendencies generated using a large database. Their overall approach is not new, but the probability curve (Figure 3b) and related maps could be useful to local managers, planners, and government officials charged with the formulation of adaptive management strategies to address marsh vulnerability to future RSLR.

I have great respect for many of the authors, but this manuscript needs to be completely rewritten and rejected in its present form. The probability curve and related maps are the most significant aspect of the manuscript, but these concepts are buried in the body of a poorly organized and confusing text. The manuscript should be rewritten with a focus on this aspect of the investigation. As submitted, it does not warrant publication in Nature Communications. Details in support of this recommendation are provided in the attached Supplemental File - Detailed Review.

Supplemental File: Detailed Review¹

Text

24. Ecologic and economic concepts never mentioned in body of text. Remove from abstract or include in body.

25. This is confusing because in line 51 it is stated the use of vertical accretion rates is inappropriate.

24-35. Abstract does not reflect the content or significance of the manuscript. Needs to be rewritten.

31. No mention of RSLR; rewrite.

36. Numerous terms are used to describe marsh response to sea level change. These include collapse, convert, expand, and retreat. Then there are terms associated with 'tendencies' including increase, decrease, transgressive, regressive, positive, and negative. Develop and use a simple and consistent terminology.

40. Published simulations are no more uncertain than this manuscript (c.f. Line 152, 159, and paragraph 164). Reword to be more specific regarding how this study is more certain than others.

44. Authors are not addressing the disparities of previous vulnerability simulations, but instead provide a new simulation for Great Britain.

44 – 53. The differing outcomes of marsh vulnerability (or resiliency) assessments are not necessarily problematic as these often reflect site-specific differences in the physical (i.e., inorganic sediment supply) and biological (i.e., above and below ground organic matter production and accumulation) setting. Just because the outcomes are different doesn't mean they are wrong, nor therefore that 'more direct measures' are needed. I don't see how the methods, conclusions, and limitations of this manuscript – as submitted - are any better than those to date. Rewrite introduction to be more specific regarding the problem being addressed and the unique outcome of this investigation. Emphasis should be placed on the methods, outcome and utility of tendency probabilities.

55. This is simply not true as there are hundreds of studies that have quantified marsh sedimentation and stratigraphy in the context of past, present, and future sea level rise. If this is not the case in Great Britain, rephrase.

56. Reference to Figure 1b is not preceded by reference to Figure 1a. Re-label. Many of the exhibit numbers in the text and supplemental are incorrectly labeled, as noted throughout this review.

¹Line numbers from Adobe file are beginning of sentence to which comments apply

55. First, move Figure 1a to follow '54 regions' to reduce some of the confusion since it does not contain the Great British (Britain?) Holocene RSL database. Even after getting through that, I'm still confused. If Figure 1b is illustrating *relative* sea level rise over time, doesn't it include the effect of glacial isostatic adjustment? Revise text to clarify.

58. Here the authors use the term 'histories'; at other locations 'change'. Be consistent. The persistent use of a broad range of terms to describe one concept is a problem throughout the text.

62. What's the significance of the differences between the three models and how does this supplemental data better inform the reader? Also, Supplemental Table 1 not referenced prior to Table 2. Assume Supplemental Table x, SOM Table x, and Table Sx are synonymous? Align terminology.

62-72. Define early, middle, and late Holocene to help the reader follow; perhaps show intervals in Figure 1b. *Center* of LGM and *central* Great Britain are similar; perhaps change one or the other term. Replace 'small or minor' Holocene 'highstand' with an interval of time. What is a 'typical' pattern of RSL change. Revise to clarify. Mid-Holocene missing in discussion of southern coasts.

73-86. This paragraph is unnecessarily confusing. For example, the following terms are used in association with a positive tendency: increase, transgressive, retreat. Decrease, regressive, and expanded with negative tendency. See also line 35 comments above. The concept of transgressive and regressive coastal facies succession in response to changing rates of RSLR is not that complicated. Rewrite to simplify.

75. Transgressive contacts are not always gradual. For example, an event driven overwash fan creates a sharp contact in an instant. Nor are they always conformable. They could be erosional and the resulting stratigraphic record incomplete. Rewrite. This of course would complicate any attempt to reconstruct the sequence of events and associated effects of SLR within a core(s). See for example SF2d sediment succession at 2 OD (what ever that means) and east (?; to the right) of core 95/3 (sand overlying peat).

76. Tidal marsh and mudflat are not lithologies. They are depositional environments. Change.

78. I think the authors mean they only used positive (transgressive) and negative (regressive) contacts (aka index points). Rewrite.

81. Repetitive and again confusing terminology.

84. Retreated (aka positive tendency, transgressive, etc.)? But authors also identified progradational (i.e., negative, regressive, etc.) contacts. What bias is being avoided? Your approach is simply different. How is it better?

88. The x-axis intervals are 0.5 mm/yr. So, the tendencies have to be described accordingly (i.e., -0.5 – 10, not -0.5 – 9.7). Why reference to SF1?

90. What modal rate?

91. Is modal class different than modal rate? And of course, the mode reflects the distribution of frequencies. What is the point?

93. How was this interval identified in Figure 2 (not 1b) – visually?

94. This statement does not follow the preceding sentence as the rate bins are lower (i.e., -0.5) and higher (i.e., 8.0) than 0 and +3 mm/yr.

95 – 102. Redundant concepts and confusing terminology.

98. I thought the authors were not including ‘index points that record marshes simply keeping pace with RSLR’ (line 80)? Rewrite.

102. I’m assuming the authors mean tendencies associated with the Holocene data set can be used to forecast marsh response to future SLR? Rewrite.

104. The probability curve is the most significant aspect of this manuscript. The text should be rewritten, as above and below, to focus on this aspect of the investigation.

116. So tendencies based upon Holocene geological record are consistent with modern observations (model validation) and therefore appropriate to forecast marsh response to future SLR. These observations are important, but not new.

125. Delete this paragraph.

134, Not SOM Table 1; perhaps Table S3?

135. Why was the RCP 8.5 scenario or the Tilbury location (probability data point with black boarder on SF3 and 4 and not described in either caption) selected? Why now introduce Common (Current) Era?

138-149. References in support of lag time should be included. Introducing emission reduction (mitigation) concepts in this section is inappropriate and disjointed relative to section heading. Also assumes reduction measures will be successfully implemented. What is ‘clear about the influence of mitigation by 2100 CE’? Static-equilibrium fingerprint, ocean dynamics and compaction; how are these related to the previous statements in this paragraph or the sentences thereafter? Text in this section seems to infer the past is not a key to the future and therefor this entire exercise is for naught. This entire paragraph is very confusing; what is the point? Paragraph should be completely rewritten to stay on topic (marsh response to future SLR).

150. Why only RCP8.5 data in Table S3 if RCP2.6 pathways are also being discussed?

154. What does ‘integrating over the uncertainties’ mean?

155. Tendencies shown in Figure 4 (gradients of red) are difficult to distinguish. Perhaps use numbers?

150-170. This section is really hard to follow. That said, having assigned probabilities to marsh tendencies in response to future SLR is very significant. It is the most important element of this manuscript and should be the focus as the results would be helpful to managers, planners, and government officials trying to develop adaptive management plan to address the consequences of RSLR. What is a coastal realignment initiative? The last sentence is unsupported by the authors' data (as they admit) and should be deleted. As written, the last two paragraphs are confusing, poorly written, and lack emphasis on the major point of this manuscript.

Figure Captions

Fig. 1. Observation: relationship between three (positive, neutral, negative) tendencies and isostasy (blue, black, red) is obvious but not well developed in body of text.

Fig. 4. Note significance of data point with black boarder.

Methods

Did not review as the main body of the manuscript needs to be completely rewritten. Review of supplemental material is warranted only after a clear and concise manuscript is prepared.

Supplementary Information

5. SOM Figure 2. Introduce regressive before transgressive (oldest first and consistent with subsequent order of text).

26. DNR

52. The order of names should be the same in Table S1 and S2.

Reviewer #2 (Remarks to the Author):

Summary

Goal of the paper: assess the (sea-level rise) limits to marsh vulnerability

Method: Compare reconstructions of tidal marshes during the Holocene around the UK to past modelled rates of sea-level change, which (spatially and temporally) vary significantly due to GIA.

Findings: At rates >7.3 mm/yr, marshes are 9 times more likely to retreat than to expand, which suggests that under high emission scenarios with larger projected sea-level rise rates, salt marshes might not be able to survive.

I enjoyed reading the paper. The methodology used seems like a good approach, as it implicitly includes a dynamical feedback and avoids the assumptions that are made in static models. I should mention though that my specialty is in sea-level change and not in marsh dynamics, so for me it is hard to judge how new this approach is and how it fits in the field. I have provided some questions and comments below.

Review comments

L63-73/Figure 1B: when using 'sea-level index points', I expected to see reconstructed/observed RSL from the various locations, corrected for GIA using the Bradley model. But judging from the lines in the graph, this is only the outcome of the GIA model at the location of the 54 regions – it looks way too perfect to be based on actual observations. Are we looking at model predictions or reconstructed observations?

I wonder: is there a risk for a circular argument when using the GIA model to compute RSL rates at the index point locations – i.e. is the GIA model completely independent from the index points used or is the model actually forced using the same information?

L81: The authors have made the choice not to include index points from marshes that keep pace with sea-level rise. I am wondering if this poses a risk for under/overstating the vulnerability of the marshes? I mean: we do not get to see if there are marshes that have kept pace with certain higher sea-level rates. By excluding these 'stable' marshes, the conclusion of 'widespread concern over their ability to survive' (l.34) only holds for those marshes that are already expanding or retreating. By leaving out the stable marshes, the conclusions might ignore that there is a certain percentage of the marshes that perhaps could cope quite well?

Might I suggest to combine figures 2 and 3 (i.e. make 1 figure with 3 panels?), since figure 3a is directly derived from figure 2? It would make it immediately clear that both figures (2 and 3a) are two ways of looking at the same data. I do think it would be good to keep both in the main text, as these two ways of showing the data give valuable information.

L121 'Impossible' seems too strong a statement here, given that there is a probability of ~90% for retreat.

L153: what is the reason for switching from emphasizing 90% probabilities (Fig 3) to 80% probabilities (Fig 4)?

For the projections, there are less tide gauge stations (41) than index point regions (54). It is not explained in the text how these two are connected, and not quite clear what the conclusions around the projected changes are based on: on the 54 index points or the index points at tide gauges alone?

“Predicting marsh vulnerability to sea-level rise using Holocene relative sea-level data”

Responses to reviewer comments

Reviewer #1

24. Ecologic and economic concepts never mentioned in body of text. Remove from abstract or include in body.

25. This is confusing because in line 51 it is stated the use of vertical accretion rates is inappropriate.

24-35. Abstract does not reflect the content or significance of the manuscript. Needs to be rewritten.

31. No mention of RSLR; rewrite.

Reply: We have rewritten the 50% of the abstract. We have removed the ecological and economic concepts and included RSLR in the first sentence. We have emphasized the significance of the paper in the last two sentences.

36. Numerous terms are used to describe marsh response to sea level change. These include collapse, convert, expand, and retreat. Then there are terms associated with ‘tendencies’ including increase, decrease, transgressive, regressive, positive, and negative. Develop and use a simple and consistent terminology.

Reply: We apologize for any confusion. We have reduced the numerous terms used to explain marsh response. These have been simplified to transgressive contacts, tidal marsh retreat and positive tendency, regressive contacts, tidal marsh expansion and negative tendency.

40. Published simulations are no more uncertain than this manuscript (c.f. Line 152, 159, and paragraph 164). Reword to be more specific regarding how this study is more certain than others.

44. Authors are not addressing the disparities of previous vulnerability simulations, but instead provide a new simulation for Great Britain.

44 – 53. The differing outcomes of marsh vulnerability (or resiliency) assessments are not necessarily problematic as these often reflect site-specific differences in the physical (i.e., inorganic sediment supply) and biological (i.e., above and below ground organic matter production and accumulation) setting. Just because the outcomes are different doesn’t mean they are wrong, nor therefore that ‘more direct measures’ are needed. I don’t see how the methods, conclusions, and limitations of this manuscript – as submitted - are any better than those to date. Rewrite introduction to be more specific regarding the problem being addressed and the unique outcome of this investigation. Emphasis should be placed on the methods, outcome and utility of tendency probabilities.

55. This is simply not true as there are hundreds of studies that have quantified marsh sedimentation and stratigraphy in the context of past, present, and future sea level rise. If this is not the case in Great Britain, rephrase.

Reply: We edited the second paragraph of the introduction section to state that our paper offer an alternative rather than superior estimate of marsh vulnerability. We inserted text and references (lines 41-61) regarding site specific differences.

In the restructuring of the paper, we created a 3rd paragraph of the introduction (lines 62-71) which emphasizes that it is a case study of Great Britain (and why this region was chosen), the methods and the outcomes of the probability distributions. We rephrase the section that suggests this is the first study of marsh sedimentation in Great Britain.

56. Reference to Figure 1b is not preceded by reference to Figure 1a. Re-label. Many of the exhibit numbers in the text and supplemental are incorrectly labeled, as noted throughout this review.

Reply: We apologize for these errors. We have corrected all errors associated with the order and nomenclature for figures and tables.

55. First, move Figure 1a to follow '54 regions' to reduce some of the confusion since it does not contain the Great British (Britain?) Holocene RSL database. Even after getting through that, I'm still confused. If Figure 1b is illustrating *relative* sea level rise over time, doesn't it include the effect of glacial isostatic adjustment? Revise text to clarify.

Reply: We have reworked the main text (lines 83, 89-92) and figure caption to clearly state Figure 1B are GIA model predictions

58. Here the authors use the term 'histories'; at other locations 'change'. Be consistent. The persistent use of a broad range of terms to describe one concept is a problem throughout the text.

Reply: We have endeavored to decrease the number of terms throughout the manuscript. For example sea-level history is replaced with sea-level change throughout

62. What's the significance of the differences between the three models and how does this\ supplemental data better inform the reader? Also, Supplemental Table 1 not referenced prior to Table 2. Assume Supplemental Table x, SOM Table x, and Table Sx are synonymous? Align terminology.

Reply: We only use one model in the main text so we have removed the section in the supplemental regarding the other two models to reduce confusion. We have corrected all errors associated with the order and nomenclature for supplementary figures and tables.

62-72. Define early, middle, and late Holocene to help the reader follow; perhaps show intervals in Figure 1b. *Center* of LGM and *central* Great Britain are similar; perhaps change one or the other term. Replace 'small or minor' Holocene 'highstand' with an interval of time. What is a 'typical' pattern of RSL change. Revise to clarify. Mid-Holocene missing in discussion of southern coasts.

Reply: We have corrected the text as suggested.

73-86. This paragraph is unnecessarily confusing. For example, the following terms are used in association with a positive tendency: increase, transgressive, retreat. Decrease, regressive, and expanded with negative tendency. See also line 35 comments above. The concept of transgressive and regressive coastal facies succession in response to changing rates of RSLR is not that complicated. Rewrite to simplify.

Reply: We have deleted this paragraph. The text regarding contacts has been rewritten and is found in the introduction paragraph 4 (lines 74-82).

75. Transgressive contacts are not always gradual. For example, an event driven overwash fan creates a sharp contact in an instant. Nor are they always conformable. They could be erosional and the resulting stratigraphic record incomplete. Rewrite. This of course would complicate any attempt to reconstruct the sequence of events and associated effects of SLR within a core(s). See for example SF2d sediment succession at 2 OD (what ever that means) and east (?; to the right) of core 95/3 (sand overlying peat).

Reply: The database only includes transgressive and regressive contacts where the change in lithology was gradations. This was stated in the original methods section. But to avoid confusion we have inserted gradual on line 107. OD is Ordnance Datum. We have explained this abbreviation in caption of Supplementary Figure 1.

76. Tidal marsh and mudflat are not lithologies. They are depositional environments. Change.

Reply: corrected

78. I think the authors mean they only used positive (transgressive) and negative (regressive) contacts (aka index points). Rewrite.

81. Repetitive and again confusing terminology.

84. Retreated (aka positive tendency, transgressive, etc.)? But authors also identified progradational (i.e., negative, regressive, etc.) contacts. What bias is being avoided? Your approach is simply different. How is it better?

Reply: We have corrected the text to simplify terminology and state that our approach is different.

88. The x-axis intervals are 0.5 mm/yr. So, the tendencies have to be described accordingly (i.e., -0.5 – 10, not -0.5 – 9.7). Why reference to SF1?

Reply: corrected descriptions of terminology and removed reference to supplemental figure.

90. What modal rate?

91. Is modal class different than modal rate? And of course, the mode reflects the distribution of frequencies. What is the point?

93. How was this interval identified in Figure 2 (not 1b) – visually?

94. This statement does not follow the preceding sentence as the rate bins are lower (i.e., -0.5) and higher (i.e., 8.0) than 0 and +3 mm/yr.

Reply: We have deleted this section to simplify the results.

95 – 102. Redundant concepts and confusing terminology.

98. I thought the authors were not including ‘index points that record marshes simply keeping pace with RSLR’ (line 80)? Rewrite.

102. I’m assuming the authors mean tendencies associated with the Holocene data set can be used to forecast marsh response to future SLR? Rewrite.

Reply: We have simplified this section including deleting line 102.

104. The probability curve is the most significant aspect of this manuscript. The text should be rewritten, as above and below, to focus on this aspect of the investigation.

Reply: We have rewritten portions of the manuscript and restructured the text. The statistical model now has its own section (lines 116-140).

116. So tendencies based upon Holocene geological record are consistent with modern observations (model validation) and therefore appropriate to forecast marsh response to future SLR. These observations are important, but not new.

Reply: Corrected.

125. Delete this paragraph.

Reply: The paragraph has been rewritten rather than deleted to introduce the section on Sea-level projections for Great Britain.

134, Not SOM Table 1; perhaps Table S3?

Reply: corrected.

135. Why was the RCP 8.5 scenario or the Tilbury location (probability data point with black border on SF3 and 4 and not described in either caption) selected? Why now introduce Common (Current) Era?

Reply: In the new paragraph introducing this section we have described that GIA is the most important process for spatial variability (lines 144-145). Tilbury is from an area of GIA subsidence. We have changed the figure captions to state Tilbury. We have removed reference to CE.

138-149. References in support of lag time should be included. Introducing emission reduction (mitigation) concepts in this section is inappropriate and disjointed relative to section heading. Also assumes reduction measures will be successfully implemented. What is ‘clear about the influence of

mitigation by 2100 CE'? Static-equilibrium fingerprint, ocean dynamics and compaction; how are these related to the previous statements in this paragraph or the sentences thereafter? Text in this section seems to infer the past is not a key to the future and therefore this entire exercise is for naught. This entire paragraph is very confusing; what is the point? Paragraph should be completely rewritten to stay on topic (marsh response to future SLR).

Reply: We have added a reference regarding lag times, and removed text relating to mitigation strategies. We have split the paragraph to improve the readability.

150. Why only RCP8.5 data in Table S3 if RCP2.6 pathways are also being discussed?

Reply: Table S4 in the original (now S2) described RCP 2.6.

154. What does 'integrating over the uncertainties' mean?

Reply: Deleted.

155. Tendencies shown in Figure 4 (gradients of red) are difficult to distinguish. Perhaps use numbers?

Reply: We don't think a different coloring scheme (e.g. red/blue) is appropriate - it will create a divide at a completely arbitrary threshold. We go from near-white at 50% probability to dark red at 100%. To improve clarity we have outlined the circles.

150-170. This section is really hard to follow. That said, having assigned probabilities to marsh tendencies in response to future SLR is very significant. It is the most important element of this manuscript and should be the focus as the results would be helpful to managers, planners, and government officials trying to develop adaptive management plan to address the consequences of RSLR. What is a coastal realignment initiative? The last sentence is unsupported by the authors' data (as they admit) and should be deleted. As written, the last two paragraphs are confusing, poorly written, and lack emphasis on the major point of this manuscript.

Reply: We have added addition sentence to clearly state the importance of the maps of timescales. We have rewritten the last paragraph of the manuscript including deleting the last sentence.

Figure Captions

Fig. 1. Observation: relationship between three (positive, neutral, negative) tendencies and isostasy (blue, black, red) is obvious but not well developed in body of text.

Reply: We have inserted lines 90-92 in main text to link to the relations between color.

Fig. 4. Note significance of data point with black boarder.

Reply: Corrected.

Supplementary Information

5. SOM Figure 2. Introduce regressive before transgressive (oldest first and consistent with subsequent order of text).

Reply: Corrected.

52. The order of names should be the same in Table S1 and S2.

Reply: Order of S1 and S2 match.

Reviewer #2

L63-73/Figure 1B: when using 'sea-level index points', I expected to see reconstructed/observed RSL from the various locations, corrected for GIA using the Bradley model. But judging from the lines in the graph, this is only the outcome of the GIA model at the location of the 54 regions – it looks way too

perfect to be based on actual observations. Are we looking at model predictions or reconstructed observations?

Reply: We have reworked the main text (lines 83, 89-92) and figure caption to clearly state Figure 1B are GIA model predictions.

I wonder: is there a risk for a circular argument when using the GIA model to compute RSL rates at the index point locations – i.e. is the GIA model completely independent from the index points used or is the model actually forced using the same information?

Reply: We employed the GIA model Bradley et al. (2011) estimate rates of relative sea-level rise. The GIA was constrained using a combination of relative sea level data from both near and far-field sites, continuous GPS data and geomorphological field data. Thus, the dataset used to constrain the GIA model was much bigger than just the transgressive and regressive contacts used in our paper to generate tendencies. We have edited the methods section to illustrate the data used to calibrate the GIA model (lines 343-348).

L81: The authors have made the choice not to include index points from marshes that keep pace with sea-level rise. I am wondering if this poses a risk for under/overstating the vulnerability of the marshes? I mean: we do not get to see if there are marshes that have kept pace with certain higher sea-level rates. By excluding these ‘stable’ marshes, the conclusion of ‘widespread concern over their ability to survive’ (L34) only holds for those marshes that are already expanding or retreating. By leaving out the stable marshes, the conclusions might ignore that there is a certain percentage of the marshes that perhaps could cope quite well?

Reply: We searched the database and found index points (n = 19) from tidal marsh peat, overlain by tidal flat deposits, but not directly from the transgressive contact and with no evidence of an increasing marine influence in either the lithology or microfossil assemblages. We classified these index points as no tendency, and they indicate the marsh is stable and keeping pace with RSLR. These index points are now included in the analysis. The conclusions of our manuscript remain unchanged, but are now more robust.

Might I suggest to combine figures 2 and 3 (i.e. make 1 figure with 3 panels?), since figure 3a is directly derived from figure 2? It would make it immediately clear that both figures (2 and 3a) are two ways of looking at the same data. I do think it would be good to keep both in the main text, as these two ways of showing the data give valuable information.

Reply: We have combined figures 2 and 3.

L121 ‘Impossible’ seems too strong a statement here, given that there is a probability of ~90% for retreat.

Reply: Replaced with very likely.

L153: what is the reason for switching from emphasizing 90% probabilities (Fig 3) to 80% probabilities (Fig 4)?

Reply: We switched to 80% in Figure 4 because this best illustrated the varying timescale of marsh retreats.

REVIEWERS' COMMENTS:

Reviewer #1 (Remarks to the Author):

The revised manuscript is a significant improvement over the initial version and the points raised in the previous round of review have been satisfactorily addressed. Minor comments are provided as an attachment to this review. I recommend acceptance after those comments are addressed.

Supplemental File: Detailed Review

Text

64. List RSLR values in same sequence as tidal marsh response here and at other locations throughout text (c.f., line 111). i.e., retreated (=15.2 mm/yr)—~~static~~, OR expanded (= - 7.7 mm/yr). Delete static as you provide no RSLR value here.

113. Figure 2a shows only negative tendency between -1.5 (not -0.5 as -1.0 includes 'no tendency value') and -5.5.

167. Figure 2c label '7.1 mm/yr' should be associated with vertical dashed line, not horizontal as that value is a probability.

287. Figure 3. What is the significance of the black dots within Islay and Tibury location circles?

Figures

Supplemental Figure 1. If possible, include transect location in 1b. 1c. Note Warkwoth location label line does not terminate on location circle as does Amble location label line. 1e ~~95/3~~. Remove 95/3.

Tables

Supplementary Table 1. Any reason RSL rise is listed in cm while Rate of RSL is in mm?

Reviewer #2 (Remarks to the Author):

Thanks to the authors for considering my previous points. I don't have any major comments that would impact the results, but I do find some unclarities remain, as indicated in the minor comments below. I think solving these would help the reader a great deal.

There are some questions arising on the rewritten abstract:

- Line 22: future sea-level rise is not the only reason that tidal marshes are vulnerable ecosystems, right? I suggest replacing 'because they' by 'which'
- Line 24: The time period that is covered by the sea-level index points is never mentioned in the entire paper. A question along the same lines was asked by reviewer 1 in their comment on l62-72 which is supposedly corrected, but I don't see this in the text (current L96-98).

Perhaps for the supplement: how many points are available at each period in time, is there a homogeneous temporal distribution?

- L 27: because of the spatially and/or temporally varying glacial isostatic response?
- Line 27 & L90-117: The RSLR rates vary between -7.7 and 15.2 mm/yr, yet Figure 2 shows only rates between -5.5 and 10 mm/yr. Is this for the reason mentioned in L 109-111? If so, this could be clarified by ending the sentence at 111 with "reducing the range of RSLR rates to -5.5-10 mm/yr".
- L28. 'indicate' (minus -d)
- L34: 'in those regions.'

L68: define 'historical', as holocene is also historical. "20th century"?

L91: How are the 54 regions defined; is this based on a certain distance from a certain point, on a certain area, or are they connected to a tide gauge location?

L 111: positive, negative, and no tendency = 3; yet only 2 ranges are given before 'respectively'.

L146 vs 154 & Supplemental Table 1/2: L154/ST1/2 is about tide gauges, but the text in 146 talks about projections for tidal marsh locations. It is unclear where and why the switch from 54 Holocene regions/ tidal marshes and to tide gauges happens. This is not explained. Are these tide gauge locations in (or close to) tidal marsh (regions)? Are the tidal marsh projections at the 54 Holocene locations – then how do the tide gauges in ST1/2 come in?

Throughout: change mm/yr. -> mm/yr

“Predicting marsh vulnerability to sea-level rise using Holocene relative sea-level data”

Responses to reviewer comments

Reviewer #1

64. List RSLR values in same sequence as tidal marsh response here and at other locations throughout text (c.f., line 111). i.e., retreated (=15.2 mm/yr) static, OR expanded (= - 7.7 mm/yr). Delete static as you provide no RSLR value here.

Reply: We have listed values in the correct order. We now provide RSLR rates for static so it remains in the text.

113. Figure 2a shows only negative tendency between -1.5 (not -0.5 as -1.0 includes ‘no tendency value’) and -5.5.

Reply: corrected.

167. Figure 2c label ‘7.1 mm/yr’ should be associated with vertical dashed line, not horizontal as that value is a probability.

Reply: corrected.

287. Figure 3. What is the significance of the black dots within Islay and Tibury location circles?

Reply: to identify Islay and Thames. Caption has been changed to make this clear

Supplemental Figure 1. If possible, include transect location in 1b. 1c. Note Warkwoth location label line does not terminate on location circle as does Amble location label line. 1e 95/3. Remove 95/3.

Reply: Location labels have been moved. We choose to keep the core label 95/3 as it links panel D and E, and the caption.

Supplementary Table 1. Any reason RSL rise is listed in cm while Rate of RSL is in mm?

Reply: Because it the common terminology when dealing with prediction of sea-level rise.

Reviewer #2

Line 22: future sea-level rise is not the only reason that tidal marshes are vulnerable ecosystems, right? I suggest replacing ‘because they’ by ‘which’

Reply: We have added the suggested text.

Line 24: The time period that is covered by the sea-level index points is never mentioned in the entire paper. A question along the same lines was asked by reviewer 1 in their comment on 162-72 which is supposedly corrected, but I don’t see this in the text (current L96-98). Perhaps for the supplement: how many points are available at each period in time, is there a homogeneous temporal distribution?

Reply: We have added a paragraph to the Methods to describe the temporal distribution (lines 213-217). We have produced a new figure (Supplementary figure 1).

L 27: because of the spatially and/or temporally varying glacial isostatic response?

Reply: Because of restriction on word count for the abstract this was not added.

Line 27 & L90-117: The RSLR rates vary between -7.7 and 15.2 mm/yr, yet Figure 2 shows only rates between -5.5 and 10 mm/yr. Is this for the reason mentioned in L 109-111? If so, this could be clarified by ending the sentence at 111 with “reducing the range of RSLR rates to -5.5-10 mm/yr”.

Reply: We have added the suggested text

L28: ‘indicate’ (minus -d)

Reply: corrected.

L34: ‘in those regions.’

Reply: corrected.

L68: define ‘historical’, as Holocene is also historical. “20th century”?

Reply: Replaced historical with 20th and 21st century.

L91: How are the 54 regions defined; is this based on a certain distance from a certain point, on a certain area, or are they connected to a tide gauge location?

Reply: We added text to the Methods (lines 194-195) to explain the definition of the regions is based on the availability of data and distance to the British-Irish ice sheet.

L 111: positive, negative, and no tendency = 3; yet only 2 ranges are given before ‘respectively’.

Reply: corrected.

L146 vs 154 & Supplemental Table 1/2: L154/ST1/2 is about tide gauges, but the text in 146 talks about projections for tidal marsh locations. It is unclear where and why the switch from 54 Holocene regions/ tidal marshes and to tide gauges happens. This is not explained. Are these tide gauge locations in (or close to) tidal marsh (regions)? Are the tidal marsh projections at the 54 Holocene locations – then how do the tide gauges in ST1/2 come in?

Reply: We added text to the Methods (lines 284-285) to explain the prediction are made for tide gauges that are located close to or within marsh regions.

Throughout: change mm/yr. -> mm/yr

Reply: corrected.